# Broad dengue neutralization in mosquitoes expressing an engineered antibody

**Anna Buchman**[1☯], **Stephanie Gamez**[1☯], **Ming Li**[1], **Igor Antoshechkin**[2], **Hsing-Han Li**[3,4,5], **Hsin-Wei Wang**[4,5], **Chun-Hong Chen**[4,5], **Melissa J. Klein**[6], **Jean-Bernard Duchemin**[6¤], **James E. Crowe, Jr.**[7,8], **Prasad N. Paradkar**[6]*, **Omar S. Akbari**[1,9]*

1 Section of Cell and Developmental Biology, University of California, San Diego, La Jolla, California, United States of America, 2 Division of Biology and Biological Engineering, California Institute of Technology, Pasadena, California, United States of America, 3 Institute of Biotechnology, National Tsing Hua University, Hsinchu, Taiwan, 4 National Institute of Infectious Diseases and Vaccinology, National Health Research Institutes, Zhunan, Taiwan, 5 National Mosquito-Borne Diseases Control Research Center, National Health Research Institutes, Zhunan, Taiwan, 6 CSIRO Health and Biosecurity, Australian Animal Health Laboratory, Geelong, VIC, Australia, 7 Vanderbilt Vaccine Center, Vanderbilt University Medical Center, Nashville, Tennessee, United States of America, 8 Departments of Pediatrics, Pathology, Microbiology and Immunology, Vanderbilt University Medical Center, Nashville, Tennessee, United States of America, 9 Tata Institute for Genetics and Society-UCSD, La Jolla, California, United States of America

☯ These authors contributed equally to this work.
¤ Current address: Institut Pasteur de la Guyane, Cayenne, French Guiana
* Prasad.Paradkar@csiro.au (PNP); oakbari@ucsd.edu (OSA)

**Data Availability Statement:** All relevant data are within the manuscript and its Supporting Information files

## Abstract

With dengue virus (DENV) becoming endemic in tropical and subtropical regions worldwide, there is a pressing global demand for effective strategies to control the mosquitoes that spread this disease. Recent advances in genetic engineering technologies have made it possible to create mosquitoes with reduced vector competence, limiting their ability to acquire and transmit pathogens. Here we describe the development of *Aedes aegypti* mosquitoes synthetically engineered to impede vector competence to DENV. These mosquitoes express a gene encoding an engineered single-chain variable fragment derived from a broadly neutralizing DENV human monoclonal antibody and have significantly reduced viral infection, dissemination, and transmission rates for all four major antigenically distinct DENV serotypes. Importantly, this is the first engineered approach that targets all DENV serotypes, which is crucial for effective disease suppression. These results provide a compelling route for developing effective genetic-based DENV control strategies, which could be extended to curtail other arboviruses.

## Author summary

With limited success of traditional vector control methods to curb dengue infections and more than half of the world's population still at risk, there is a need for novel strategies to reduce its impact on public health. Recent advances in genetic technologies has allowed for precise modifications of mosquito genome to make them resistant to infections, thus breaking the transmission cycle. Here we generated engineered *Ae. aegypti* mosquitoes

**Funding:** This work was supported in part by a Defense Advanced Research Project Agency (DARPA) Safe Genes Program Grant (HR0011-17-2- 0047) awarded to O.S.A. and a NIH Exploratory/ Developmental Research Grant Award (1R21AI123937) awarded to O.S.A and CSIRO internal funding to P.N.P. The funders had no role in study design, data collection, analysis, the decision to publish, nor the preparation of the manuscript

**Competing interests:** O.S.A is a founder and serves on the scientific advisory board for Agragene. J.E.C. has served as a consultant for Takeda Vaccines, Sanofi Pasteur, Pfizer, and Novavax, is on the Scientific Advisory Boards of CompuVax, GigaGen, Meissa Vaccines, and is the Founder of IDBiologics, Inc. All other authors declare no competing financial interests.

efficiently expressing a DENV-targeting single-chain variable fragment (scFv) derived from a previously characterized broadly neutralizing human antibody, which blocked infection and transmission in these mosquitoes. To our knowledge, this is the first example of an engineered transgene capable of rendering *Ae. aegypti* mosquitoes 100% refractory to all four serotypes of DENV. The engineered mosquitoes, in future, could easily be paired with a gene drive, capable of spreading the transgene throughout wild disease-transmitting mosquito populations and preventing further DENV transmission. Since a number of diverse and well-characterized antibodies exist against other arboviruses (eg chikungunya and Zika, this work also provides a proof-of-concept principle for developing similar genetic strategies for reducing the impact of these arboviruses.

## Introduction

Dengue fever is a devastating viral disease caused by several antigenically distinct dengue viral (DENV) serotypes that are all dangerous to humans but cannot be readily controlled using broad-spectrum techniques [1,2]. Transmitted by infected mosquitoes, DENV infection typically manifests as severe fever, headaches, and myalgia[3] and can advance to the life-threatening dengue hemorrhagic fever and dengue shock syndrome[4]. Global incidences of DENV and its associated economic burden have increased dramatically in recent years [5,6], with over 50% of the world's population now at risk of infection[7] and 390 million documented infections per year[6] for an estimated $40 billion in economic losses annually[8,9]. Moreover, there are currently no specific treatments nor preventive prophylactic measures [10] because the single commercially available vaccine [11] is only partially effective [12], and due to increased risk of severe dengue illness and hospitalization among certain groups, its use is prevented in many contexts [13–15]. Therefore, control of the disease-spreading mosquitoes is currently the best option for preventing DENV transmission [13].

*Aedes aegypti* [16], the main vector of DENV and other epidemiologically significant viruses such as chikungunya (CHIKV), yellow fever (YFV), and Zika (ZIKV), is a domestic endophilic mosquito[17] that has expanded its habitable range in recent decades[18] and will likely continue to spread[19]. Current control measures including the removal of standing water and the use of chemical insecticides have had limited success in reducing *Aedes* populations [20] and, thereby, incidences of DENV[21], and can instead cause insecticide resistance and behavioral changes such as a shift in biting times[22,23]. Therefore, novel vector control strategies[24], like the use of genetically modified mosquitoes to either suppress mosquito populations or render mosquitoes unable to transmit pathogens[25], are increasingly needed. For example, the development and deployment of a genetic Sterile Insect Technique (SIT)-like system termed Release of Insect Dominant Lethal (RIDL) has had some success in reducing *Aedes* mosquito populations in the wild[26,27]. Moreover, releases of mosquitoes artificially infected with the intracellular endosymbiont *Wolbachia*, which can make infected males incapable of successfully mating with uninfected females in an SIT-like manner and can inhibit mosquito infection with pathogens such as DENV and ZIKV[28,29], have also been carried out. These have been intended to either suppress mosquito populations or make them less likely to transmit pathogens, and may hold promise for reducing the incidence of disease [30,31]. However, these technologies require releases of large numbers of insects—and must be carried out on an ongoing basis for RIDL and *Wolbachia*-based SIT—for continued vector control, which is laborious and expensive.

Therefore, there has been increasing interest in the development of engineered gene-drive technologies, which are able to rapidly transmit themselves and any linked "cargo" genes, such as anti-pathogen effectors, through wild disease-transmitting populations [25,32–35] such that only a few releases of modest amounts of engineered insects could drive desirable cargo genes through wild populations, making them efficient and cost effective for vector control. To achieve disease reduction, such gene-drive systems need to be linked to useful "cargo", such as effective anti-pathogen genes, and several approaches for engineering *Ae. aegypti* resistance to DENV have been attempted. For example, one study used RNA interference by employing inverted RNA repeats to target DENV-2 in a conditional and tissue-specific manner[36,37], while another described miRNA cassettes targeting DENV-3 that reduced viral transmission rates[38]. In addition to using synthetic small RNAs, others have taken advantage of the mosquito's innate antiviral JAK/STAT pathway to increase resistance to DENV-2 and DENV-4 [39]. However, all previous approaches have been limited by their ability to target only one or two—not all four—major DENV serotypes. Because hyperendemicity of DENV in tropical areas is frequent[5] and secondary DENV infection has been linked to severe dengue disease (SDD), refractory mosquitoes should be capable of blocking all serotypes or risk being ineffective in controlling dengue epidemics. Therefore, better anti-DENV effectors are needed.

Broadly neutralizing antibodies may be especially promising as anti-DENV effector gene candidates because of their ability to neutralize antigenically diverse viruses[40]. However, while engineered monoclonal antibodies that confer resistance to *Plasmodium*, a protozoan parasite that causes malaria, have been expressed in *Anophelene* mosquitoes[41–43], none targeting a virus have been described in any mosquito species. Previously, a DENV-targeting 1C19 monoclonal antibody (MAb) was identified from a large panel of naturally occurring MAbs from human subjects following vaccination or natural infection[44]. *In vitro* studies demonstrated that this antibody neutralized viruses from all major DENV serotypes and was capable of significantly reducing viremia in a mouse model after DENV-1 and DENV-2 infection[44]. Here, we engineer *Ae. aegypti* to express a 1C19-based, broadly neutralizing, single-chain variable fragment (scFv) that is capable of neutralizing all four DENV serotypes[44]. Crucially, we demonstrate that mosquitoes expressing this anti-DENV scFv cannot be infected with or transmit any of the four DENV serotypes and have few significant fitness costs conferred by the presence of the antibody. These results provide a promising route for developing effective DENV control strategies using genetically engineered mosquitoes.

## Materials and methods

### Anti-DENV scFv design

Sequences for the 1C19 variable heavy and light chains were obtained from hybridoma cells expressing the human monoclonal antibody [44] that had been cloned biologically by flow cytometry. RNA was extracted using the RNeasy kit (Qiagen #74104), and RT-PCR amplification of antibody gene cDNAs was performed using primer sets designed to detect all human antibody variable gene sequences [44]. The sequence of the antibody cDNAs was determined by automated Sanger sequence analysis. The sequence analysis of the antibody variable gene sequences in the cDNAs was performed using the international ImMunoGeneTics information system (IMGT).

The variable regions of 1C19 were joined by a 15-amino-acid repeating glycine-serine [G(4)S]3 linker[45] to encode a scFv form of the antibody[46]. These chain regions were codon optimized for *Ae. aegypti* expression and then gene synthesized into a vector (GenScript, Piscataway, NJ). For OA984-HA, a 3' 30-amino-acid human influenza hemagglutinin (HA)

epitope tag with a G(4)S linker[47] was added to the carboxy terminus of the single chain antibody for protein expression verification.

## Plasmid assembly

To generate vector OA984 (the anti-DENV scFv-antibody transgene), several components were cloned into the *PiggyBac* plasmid pBac[3xP3-DsRed] (a kind gift from R. Harrell) using Gibson assembly/enzymatic assembly (EA) cloning [48]. First, a *Drosophila* codon-optimized tdTomato marker was amplified with primers 984.1A and 984.1B (Supplementary S2 Table for all primers) from a gene synthesized vector (GenScript, Piscataway, NJ) and cloned into a XhoI/FseI-digested pBac[3xP3-DsRed] backbone using EA cloning. The resulting plasmid was digested with AscI, and the following components were cloned in via EA cloning: the predicted *Ae. aegypti* carboxypeptidase promoter [49] amplified from *Ae. aegypti* genomic DNA using primers 984.2A and 984.2B, a GFP sequence amplified from vector pMos[3xP3-eGFP] [50] with primers 984.3A and 984.3B, and a 677-bp p10 3' untranslated region (UTR) amplified with primers 984.4A and 984.4B from vector pJFRC81-10XUAS-IVS-Syn21-GFP-p10 (Addgene plasmid #36432). The anti-DENV scFv was then subcloned into the final vector from a gene-synthesized plasmid (GenScript, Piscataway, NJ) using PmeI and PacI sites and traditional ligation cloning. Annotated plasmid sequences and plasmid DNA are available via Addgene (plasmid #120363).

To generate vector OA984-HA (anti-DENV scFv with HA-epitope tag), the G(4)S linker and HAx3 tag were amplified with primers 984B.C1 and 984B.C2 from the *ninaE*[SBP-His] vector containing these components [51] and cloned into the PacI digested OA984 backbone using EA cloning. Annotated plasmid sequences and plasmid DNA are available via Addgene (plasmid #120362). All primer sequences used to generate these plasmids are listed in S2 Table.

## Generation of transgenic mosquitoes

Germline transformations were carried out largely as described [52]. Briefly, 0–1 hr old Higgs wildtype (WT) *Ae. aegypti* pre-blastoderm embryos were injected with a mixture of vector OA984 or OA984-HA (200 ng/µL) and a source of *PiggyBac* transposase (200 ng/µL)[50]; the injected embryos were hatched in deoxygenated $H_2O$. A total of 127 surviving WT adult $G_0$ males and 115 surviving WT adult $G_0$ females were recovered after the injection. Microinjected WT $G_0$ adults were assigned to 48 pools and outcrossed to WT of the opposite sex in medium-sized cages. Larvae were fed ground fish food (TetraMin Tropical Flakes, Tetra Werke, Melle, Germany) and adults were fed with 0.3 M aqueous sucrose. Adult females were blood fed three to five days after eclosion using anesthetized mice. All animals were handled in accordance with the Guide for the Care and Use of Laboratory Animals as recommended by the National Institutes of Health, and the methods were supervised by the local Institutional Animal Care and Use Committee (IACUC). A total of 38,177 WT $G_1$s were screened. $G_1$ larvae with strong positive fluorescent signals (3xp3-tdTomato) were selected under the fluorescent stereomicroscope (Leica M165FC) and were separated into six individual groups characterized by fluorescence patterning and intensity. One single transgenic male from each group was then allowed to separately mate with 10 WT females to isolate each independent line. Three independent lines, TADV-A (vector OA984), TADV-B (vector OA984-HA), and TADV-C (vector OA984-HA) with the strongest fluorescence expression patterns were selected for further characterization. To determine whether these lines represented single chromosomal insertions, we backcrossed single individuals from each of the lines for four generations to WT stock and measured the Mendelian transmission ratios in each generation; in all cases, we

observed a 50% transmission ratio, indicating single-chromosome insertion. For one of the three lines (TADV-A), transgenic mosquitoes were inbred for at least 20 generations to generate a homozygous stock. Mosquito husbandry was performed under standard conditions as previously described [53].

## Characterization of insertion sites

To characterize the insertion site of vector OA984 or OA984-HA in transgenic mosquitoes, we adapted a previously described inverse polymerase chain reaction (iPCR) protocol [54] as follows. First, genomic DNA (gDNA) was extracted from 10 transgenic fourth instar larvae using the DNeasy Blood & Tissue Kit (Qiagen #69504) per the manufacturer's protocol. Two separate restriction digests were performed on the gDNA (at 100 ng/μL) to characterize the 5' and 3' ends of the *PiggyBac* insertion using Sau3AI (5' reaction) or HinP1I (3' reaction) restriction enzymes. A ligation step using NEB T4 DNA Ligase (NEB #M0202S) was performed on the restriction digest products to circularize digested gDNA fragments, and two subsequent rounds of PCR were carried out per ligation using the corresponding *PiggyBac* primers listed in S3 Table. The final PCR products were cleaned up using the MinElute PCR Purification Kit (Qiagen #28004) in accordance with the manufacturer's protocol and were sequenced via Sanger sequencing (Source BioScience, Nottingham, UK). To confirm the transgene insertion locus and orientation via PCR, primers were designed based on iPCR-mapped genomic regions and used in tandem with *PiggyBac* primers based on their location as listed in S3 Table. Sequencing data then was blasted to the AaegL5.0 reference genome [55]. The sequencing data was aligned with SeqManPro (DNASTAR, Madison, WI) to determine the orientation of the transgene insertion site. Analysis of the sequencing data indicated that the insertion site for TADV-A is on chromosome 2 (approximate position 310,340,476), the insertion site for TADV-B is on chromosome 2 (approximate position 301,489,980), and the insertion site for TADV-C is on chromosome 1 (approximate position 30,451,048) when aligned to the AaegL5 assembly (GenBank assembly accession: GCA_002204515.1)[56].

## Total RNA extraction, isolation, and sequencing

Total RNA was extracted from the midguts of non-blood-fed and 24-hours post-blood-fed TADV-A, TADV-B, TADV-C or WT adult females using the Ambion mirVana mRNA Isolation Kit (ThermoFisher Scientific #AM1560). Following extraction, the RNA was treated with Ambion Turbo DNase (ThermoFisher Scientific #AM2238). The RNA quality was assessed using an RNA 6000 Pico Kit for Bioanalyzer (Agilent Technologies #5067–1513) and a Nano-Drop 1000 UV-vis spectrophotometer (NanoDrop Technologies/Thermo Scientific, Wilmington, DE). mRNA was isolated using an NEBNext Poly(A) mRNA Magnetic Isolation Module (NEB #E7490), and libraries were constructed using an NEBNext Ultra II RNA Library Prep Kit for Illumina (NEB #E7770). The libraries were quantified using a Qubit dsDNA HS Kit (ThermoFisher Scientific #Q32854) and a High Sensitivity DNA Kit for Bioanalyzer (Agilent Technologies #5067–4626) and sequenced on an Illumina HiSeq2500 in single-read mode with a read length of 50 nt and sequencing depth of 30 million reads per library following the manufacturer's instructions. Reads were mapped to the *Ae. aegypti* genome (AaegL5.0) supplemented with the 1C19 cDNA sequence using STAR aligner [57], and the expression levels were determined with featureCounts [58] (S4 Table). Correlation coefficients of the transcripts-per-million (TPM) values between WT and transgenic animals were calculated in R [14] and plotted with ggplot2 (S1 Fig). Differential expression analysis between transgenic and WT sample pairs of the same feeding status (NBF or PBM) for each line using DESeq2 [59] identified no significantly changed genes (padj < 0.05) for all six comparisons (data not

shown). To increase the sensitivity of the assay, two factor analysis using both NBF and PBM samples per line with design = ~ feeding + genotype was also performed and identified a number of differentially expressed genes for each line (S5 Table). However, expression of only nine genes was consistently altered in all three lines (S6 Table), suggesting that expression of the 1C19 scFv transgene had minimal impact on overall expression patterns of endogenous genes. All sequencing data can be accessed at NCBI SRA (study accession ID PRJNA524725).

## Western blot assays

The general western blot protocol was adapted from CSH Protocols: SDS-PAGE of Proteins [60]. Briefly, 5–7 days post eclosion, midguts from 25 non-blood-fed and 16-hour post-blood-meal heterozygous TADV-A transgenic and WT mosquitoes were dissected and collected in 1x PBS. Protein samples from dissected tissues were extracted with ice-cold radioimmunoprecipitation assay buffer (RIPA buffer; 50 mM Tris-HCl pH 7.4, 150 mM NaCl, 0.25% Na-deoxycholate, 1% NP-40, 1 mM EDTA). The protein concentration was measured using Protein Assay Dye (Bio-Rad, Cat. No#5000006) and multi-detection microplate readers (Molecular Devices, SpectraMax M2). Next, 40 μg of total protein were run on a 12% SDS-PAGE and transferred onto a 0.45 μm Immobilon-P Transfer Membrane (Merck Millipore, Cat. NO#IPVH00010). The membrane was hybridized with a custom antibody at a 1 μg/mL dilution (GenScript, Item number: U3233DA170_2) to directly recognize the 1c19 scFv peptide (26.3KDa) as well as a monoclonal antibody specific to the HA tag for lines TADV-B and C (Cell Signaling, #3724S) at a 1:1,000 dilution; these were subsequently detected by using rabbit IgG antibody (HRP) (GeneTex, Cat. No#GTX 213110–01) at a 1:10,000 dilution. Images were generated by applying the chemiluminescent HRP substrate (Millipore, Cat. No#WBKLS0500) to the blots.

## DENV infection of mosquitoes and virus determination

All experiments were performed under biosafety level 3 (BSL-3) conditions in the insectary at the Australian Animal Health Laboratory. The following DENV strains were used for all viral challenge experiments: DENV-1 (isolate ET243, GenBank EF440432), DENV-2 (isolate ET300, GenBank EF440433), DENV-3 (isolate ET209, GenBank EF440434), DENV-4 (isolate ET288, GenBank EF440435). The virus was passaged in Vero cell monolayer cultures before use for mosquito infections. WT or transgenic (confirmed by red fluorescence in the eye) mosquitoes were exposed to DENV as described previously [61]. Briefly, female mosquitoes were challenged with an infected blood meal ($TCID_{50}$ /mL) through membrane feeding using chicken blood and skin. For infection frequency and virus titer, mosquito midguts were collected at 4 dpi. For dissemination and transmission frequency, mosquito saliva, midguts, and carcasses were collected at 14 dpi. Mosquito saliva was used to determine viral titers using a $TCID_{50}$ assay on Vero cell monolayer cultures. Midguts and carcasses were used to determine the presence of viral RNA using RT-qPCR against NS5. Mosquito viral challenge, processing, saliva testing, and molecular analyses of infection and dissemination were carried out as previously described [61]. DENV infection frequency was defined by the number of midguts (day 4) found positive for viral nucleic acid. Similarly, the dissemination frequency was calculated by the number of carcasses (day 14) found positive by qPCR. Transmission frequency was defined by the number of $TCID_{50}$-positive saliva samples over the number tested. These different frequencies and average $TCID_{50}$ values were compared by the Student's two-tailed t-test.

## Confirmation of transgene zygosity

Both homozygous and heterozygous (generated by crossing out homozygous individuals to WT) mosquitoes were used for assays. To confirm the zygosity of tested transgenic mosquitoes, mosquito heads were homogenized using a bead-beater device for DNA extraction in 30 μL of extraction buffer (1x Tris-EDTA, 0.1 M EDTA, 1 M NaCl, and 2.5 μM proteinase K) and incubated at 56˚C for 5 minutes and then at 98˚C for 5 minutes. The first round of PCR was performed to test for the presence of the anti-DENV transgene using primers 991.3F1 and 1018.S19 (S3 Table). Another round of PCR was then performed using primers 1018.S19 and 1018.S21 (S3 Table) to amplify the WT insertion locus (i.e., locus lacking transgene insertion) and thus determine zygosity. For TADV-B, primer set 991.3R2 and 1018.S73 was used to amplify the anti-DENV transgene and primer set 1018.S73 and 1018.S74 were used to amplify the WT insertion site. For TADV-C, primer sets 991.3F2 and 1018.S80 and set 1018.S80 and 1018.S82 were used to amplify the anti-DENV transgene and WT insertion site, respectively. WT mosquitoes served as controls to ensure that the WT locus was successfully amplified in the genetic background. A PCR kit (ThermoFisher Scientific #F553S) with a 57˚C annealing temperature was used for all PCRs following standard protocols.

## Generation of *w*Mel *Wolbachia* line and infection assay

Eggs of *Ae. aegypti* infected with the *Wolbachia* strain *w*Mel were obtained from the World Mosquito Program (Prof. Scott O'Neill, Monash University). WT mosquitoes infected with *w*Mel were generated by crossing *w*Mel+ females with males from the WT line, and the resulting offspring were used for DENV infection experiments. At the end of the experiment, the *Wolbachia* infection status of these mosquitoes was tested using PCR with primers specific for *w*Mel detection [62] (S3 Table). The PCRs indicated the presence of *w*Mel in >90% of mosquitoes, and only results from these positive mosquitoes were used for further analysis.

## Fitness evaluation on transgenic anti-DENV mosquitoes

To determine if the anti-DENV transgene conferred a fitness cost, several fitness parameters were evaluated in TADV-A transgenic heterozygous and sibling WT mosquitoes. The evaluations of all experimental and control replicates were performed simultaneously. Insectary conditions were maintained at 28˚C and 70 to 80% in relative humidity with a 12 hr light/dark cycle. To assess the larval to pupal development time, the eggs were vacuum hatched, and the larvae were distributed into pans (50 larvae per pan) containing 2.5 L of ddH$_2$O and 0.6 mL of fish food slurry. To determine the larval to pupal development time of transgenic and WT control mosquitoes, the larvae were allowed to pupate, and pupae were collected and counted every day until no pupae were left. To assess female fertility and fecundity, 90 WT and transgenic females were mated to 20 WT males in a cage. After four days, the females were blood fed and individually transferred into plastic vials filled with water and lined with egg paper. After three days, egg papers were collected, and the eggs were counted and vacuum hatched in nine-ounce plastic cups. Starting on the fourth day, the larvae were counted every day until no larvae were present. Female fecundity refers to the number of eggs laid per female, and fertility reflects the number of eggs hatching to produce larvae. To measure male mating success, fecundity, and fertility, one TADV-A transgenic or WT male was mated to five WT females in a single mesh-covered cup filled with water and lined with egg paper. Three days post blood meal, the cups were checked for the presence of eggs, which were collected, counted, and hatched. Hatched larvae were then counted every day until no larvae were present. Male mating success was calculated as the percentage of single male outcrosses that produced larvae. Fecundity was measured as the number of eggs laid per cup; fertility was determined by the

number of hatching larvae in each cup. Finally, to assess mosquito longevity, equal numbers of male and female TADV-A transgenic or WT mosquitoes were placed in medium-sized cages (in triplicate). Mosquitoes that died were counted and removed daily until all mosquitoes had died. Statistical analyses were performed using GraphPad Prism software (GraphPad Software, La Jolla, California, USA). The means were compared using unpaired t tests with Welch's correction, with the exception of male mating success that did not use Welch's correction. The analyses of mosquito survivorship used the Mantel-Cox test. $P$ values $> 0.05$ were considered not significant.

## Results

### Generation of DENV-resistant mosquitoes

To determine whether expressing an anti-DENV antibody in mosquitoes could confer resistance to DENV, we first needed to engineer a broadly neutralizing antibody that was compatible with mosquitoes and could be expressed *in vivo* in its desired form. We chose 1C19 as our model due to its ability to cross-neutralize multiple DENV serotypes in humans[44]. As it is a human monoclonal antibody, however, it cannot be unobtrusively expressed in mosquitoes, so a new form that is both compatible with mosquitoes and maintains its neutralization capabilities had to be designed. We then choose to engineer an scFv comprising the linked variable heavy (VH) and light (VL) chains because this format removes the human-specific constant region that could impart difficulties in a mosquito and it can be expressed in one "chunk" in an organism without the need for additional *in vivo* processing. To do this, sequences for the 1C19 VH and VL chains were obtained from hybridoma cells expressing the human monoclonal antibody[44]. We then engineered a scFv comprising the VH and VL domains of 1C19 linked using a 15-amino-acid repeating glycine-serine [G(4)S]3 linker[45] that was codon-optimized for *Ae. aegypti*. We also engineered a version of this 1C19 scFv that was fused with a 3' 30-amino-acid human influenza hemagglutinin (HA) epitope tag, commonly used as a general expression tag, reasoning that it might be useful in downstream expression analyses. To conditionally drive expression of the 1C19 scFvs in the midgut of female mosquitoes following a blood meal, which would ensure 1C19 expression any time the mosquito was in contact with DENV, we used the *Ae. aegypti* carboxypeptidase (CP) promoter [49], which should induce expression in the midgut following blood ingestion (Fig 1A). (Previous findings determined that the CP promoter induces enhanced transcription of *Aedes aegypti* CPA mRNA after a blood meal and a somewhat moderate expression in sugar-fed mosquitoes [63].)The engineered anti-DENV transgenes (termed plasmid OA984 for the untagged version and plasmid OA984-HA for the HA-tagged version) also contained an eye-specific 3xP3 promoter[64], driving expression of tdTomato as a fluorescent transgenesis marker. Following the typical transgenesis procedure in mosquitoes, consisting of embryonic microinjection and $G_0$ outcrossing, multiple independent transgenic lines (n = 6) were readily identified in the $G_1$ generation via the robust expression of tdTomato fluorescence; three of the lines with the strongest marker expression (termed Transgenic Anti-DENV [TADV]-A, containing OA984; and TADV-B and C, containing OA984-HA) were selected for further experiments. We carried out inverse PCR (iPCR) on genomic DNA extracted from the transgenic mosquitoes to verify the transgene insertion site and performed backcrosses to WT for multiple generations to ensure that the transgenic lines represented single chromosomal insertions, and were able to confirm that, in all three independent lines, the anti-DENV transgenes were stably integrated into single chromosomes.

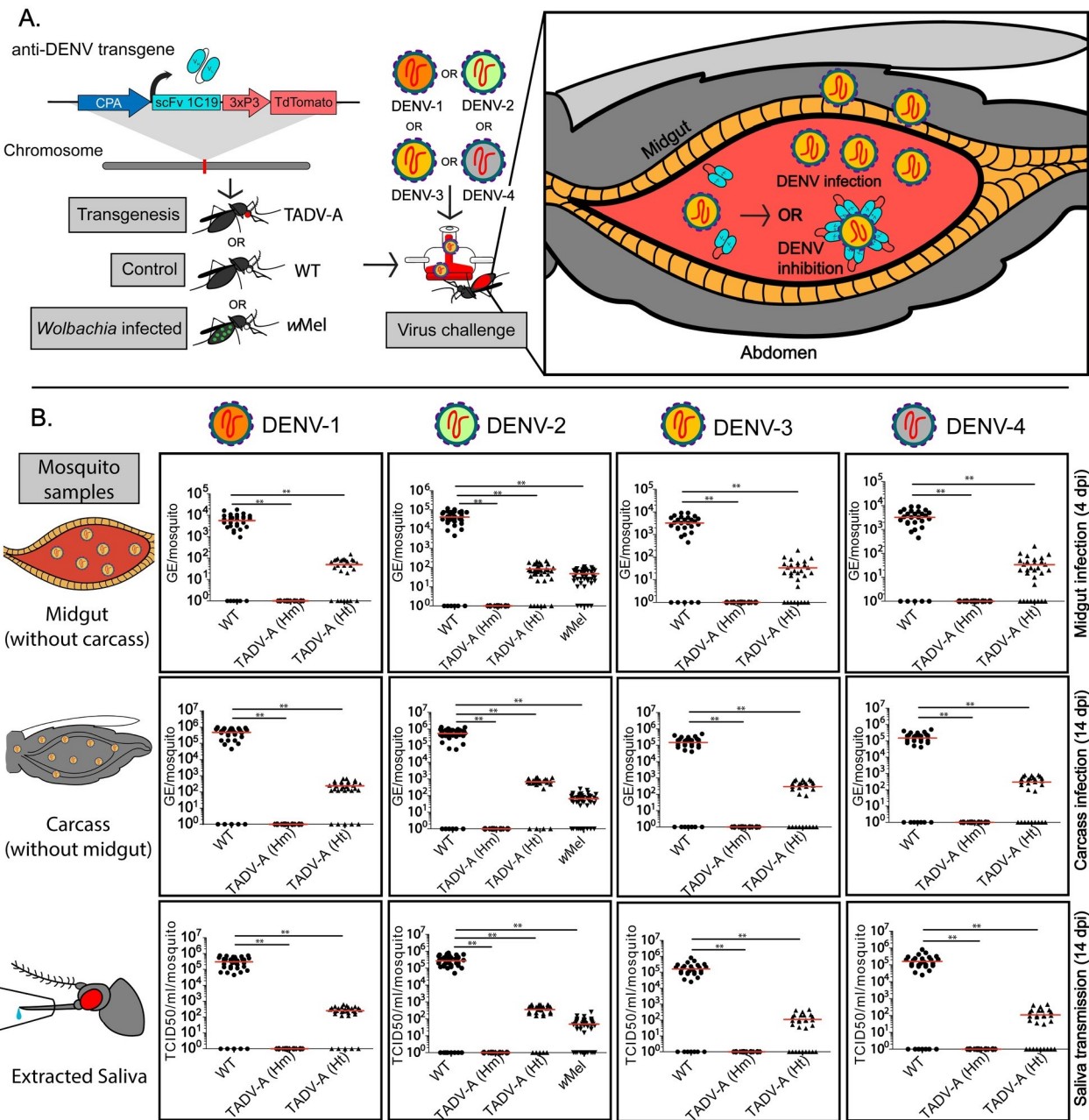

**Fig 1. Effect of anti-dengue virus (DENV) single-chain variable fragment (scFv) on DENV titers of TADV-A, *Wolbachia*-infected (*w*Mel), and wildtype (WT) mosquitoes.** (A) Schematic of experiment. TADV-A mosquitoes were generated via transgenesis with the anti-DENV construct, and TADV-A, *w*Mel, and WT mosquitoes were then challenged with a blood meal infected with one of four DENV serotypes (DENV-1, isolate ET243; DENV-2, isolate ET300; DENV-3, isolate ET209; or DENV-4, isolate ET288). After the infected blood meal enters the mosquito midgut, there are two potential outcomes: in the first (applies for all tested strains), the virus replicates and disseminates past the midgut to become transmissible; in the second (applies to TADV-A mosquitoes), the anti-DENV transgene expresses scFv antibodies in the midgut that bind to the virus and neutralize it. (B) Plots depicting viral titers. To determine if the anti-DENV transgene confers resistance to all four DENV serotypes, we determined viral titers in extracted midguts, carcasses, and saliva from WT, TADV-A (homozygous [Hm] and heterozygous [Ht]), and *w*Mel infected mosquitoes. Viral genome equivalents (GE) from mosquito midguts (at 4 days post infection [dpi]) and carcass (at 14 dpi) were determined using RT-qPCR and calculated using previously published methods. Viral titers in the saliva were determined using the median tissue culture infective dose ($TCID_{50}$) on Vero cells. For each experiment, data from three replicates is pooled. Red horizontal bars represent the mean GE/viral titer. $**p < 0.001$.

## Expression of 1C19 scFv antibody

Robust expression and processing of 1C19 scFv transcripts is required for proper neutralization of DENV, and it is important to know if such expression perturbs global gene-expression patterns, which might interfere with the fitness of the mosquito. To confirm this, we sequenced total RNA populations from dissected midgut tissues isolated from both blood-fed and non-blood-fed female Higgs wildtype (WT) or TADV-A, B, or C mosquitoes using an Illumina platform. We detected robust expression of the 1C19 scFv mRNA in both non-blood-fed and 24-hour post-blood-fed transgenic mosquitoes of all three lines, with clearly increased expression levels at 24-hours post-blood meal, while no expression was observed in the midguts of female WT mosquitoes, suggesting that expression of the 1C19 scFv antibody is transgene-dependent and blood-meal-inducible (~2.35–14.45 fold) as was intended. Importantly, while there were some changes in expression of some genes in transgenic mosquitoes when compared to WT, these represented a small fraction of the genome and, with a few exceptions, did not appear to be consistent between the three transgenic lines (S5 and S6 Tables). This suggests that the 1C19 scFv expression does not affect gene-expression patterns in a major, global way (S1 Fig, S4–S6 Tables).

To confirm the proper expression of the 1C19 scFv, we performed western blots on dissected midgut tissue from non-blood-fed and blood-fed WT and TADV-A as well as blood-fed TADV-B and TADV-C female mosquitoes using either a custom anti-1C19 scFv antibody, or an anti-HA antibody. Blot analyses revealed that the 1C19 scFv peptide was efficiently expressed following a blood meal exclusively in transgenic mosquitoes (S2 Fig). Altogether, these results suggest that the anti-DENV transgene is stably integrated into the mosquito genome and that the DENV-targeting 1C19 antibody is expressed in an appropriate context (i.e., in the midgut following a blood meal) for DENV suppression.

## Engineered mosquitoes are resistant to infection with all four DENV serotypes

To determine the functional significance of anti-DENV 1C19 scFv expression on vector competence, the DENV-2 infection rates of the three transgenic TADV lines were first compared to that of WT. To do this, adult females (WT or TADV-A, B, or C) were exposed to serotype DENV-2 (isolate ET300, Genbank EF440433) via membrane blood feeding. At 4 days post infection (dpi), midguts from blood-fed mosquitoes were dissected, and DENV RNA copies were measured using RT-qPCR. All three lines showed a significant reduction in midgut infection rate (45 to 71%) and viral RNA levels (2 to 3 log lower) compared with WT control mosquitoes (infection rate 92%) (S3 Fig; S1 Table). Since no significant difference in DENV-2 midgut infection levels was detected between the three transgenic lines, TADV-A, which exhibited the strongest antiviral phenotype (S3 Fig; S1 Table), was selected for further comprehensive characterization.

For a more detailed study of the TADV-A DENV vector competence, the effect of transgene zygosity on the infection rate was determined by exposing additional adult females (WT or TADV-A) to serotype DENV-2 and analyzing the dissected midguts at 4 dpi. Results from three biological replicates revealed that none of the TADV-A mosquitoes homozygous for the transgene (n = 35) were positive for DENV-2 infection in the midguts (Fig 1B). DENV-2 infection was detected in 85.4% (35/41) of the TADV-A mosquitoes that were heterozygous for the transgene; however, these mosquitoes had significantly ($p < 0.001$) lower (~3 $\log_{10}$) viral RNA levels ($8.20 \times 10^1$ genome equivalent [GE]) than the WT ($4.25 \times 10^4$ GE) (Fig 1B, Table 1).

To assay for viral dissemination to the rest of the mosquito body, total RNA was collected from whole TADV-A mosquito carcasses (without midguts) and dissected midguts from both

**Table 1. Anti-DENV scFv effect on DENV infection, dissemination, and transmission rates.** DENV titers in WT, heterozygous and homozygous TADV-A (TAD-V-A$_{Ht}$ and TADV-A$_{Hm}$, respectively), and $w$Mel mosquitoes following a blood meal infected with one of four DENV serotypes are shown. DENV GE from mosquito midguts (at 4 or 14 dpi) and carcasses (14 dpi) of WT, TADV-A, and $w$Mel (for DENV-2 only) mosquitoes were determined using RT-qPCR and calculated using previously published methods. Viral titers in saliva collected from WT, TADV-A, and $w$Mel mosquitoes at 14 dpi were determined using TCID$_{50}$ on Vero cells.

| | | DENV GE or viral titer on indicated dpi from specified tissue (proportion of mosquitoes with detected virus; percentage of mosquitoes with detected virus) | | | |
|---|---|---|---|---|---|
| DENV Serotype | Strain | 4 dpi | 14 dpi | | |
| | | Midgut–GE | Midgut–GE | Carcass–GE | Saliva–viral titer |
| DENV-1 | WT | $5.70 \times 10^3$ (29/35; 83%) | $3.80 \times 10^5$ (35/40; 88%) | $4.80 \times 10^5$ (35/40; 88%) | $3.04 \times 10^5$ (35/40; 88%) |
| | TADV-A$_{Ht}$ | $5.00 \times 10^1$ (20/28; 71%) | $3.70 \times 10^2$ (25/30; 83%) | $2.40 \times 10^2$ (25/30; 83%) | $2.50 \times 10^2$ (25/30; 83%) |
| | TADV-A$_{Hm}$ | $0.00 \times 10^1$ (0/28; 0%) | $0.00 \times 10^1$ (0/28; 0%) | $0.00 \times 10^1$ (0/28; 0%) | $0.00 \times 10^1$ (0/28; 0%) |
| DENV-2 | WT | $4.25 \times 10^4$ (34/40; 85%) | $4.40 \times 10^5$ (40/46; 87%) | $5.35 \times 10^5$ (40/46; 87%) | $2.70 \times 10^5$ (38/46; 83%) |
| | TADV-A$_{Ht}$ | $8.20 \times 10^1$ (35/41; 85.4%) | $3.90 \times 10^2$ (26/30; 86.6%) | $6.70 \times 10^2$ (26/30; 87%) | $3.56 \times 10^2$ (25/30; 83.3%) |
| | $w$Mel | $4.75 \times 10^1$ (43/48; 90%) | $5.10 \times 10^1$ (38/48; 79%) | $6.45 \times 10^1$ (38/48; 79%) | $4.90 \times 10^1$ (35/48; 73%) |
| | TADV-A$_{Hm}$ | $0.00 \times 10^1$ (0/35; 0%) | $0.00 \times 10^1$ (0/30; 0%) | $0.00 \times 10^1$ (0/30; 0%) | $0.00 \times 10^1$ (0/30; 0%) |
| DENV-3 | WT | $1.80 \times 10^4$ (23/30; 77%) | $2.90 \times 10^5$ (29/35; 83%) | $3.50 \times 10^5$ (29/35; 83%) | $2.90 \times 10^5$ (29/35; 83%) |
| | TADV-A$_{Ht}$ | $3.60 \times 10^1$ (22/30; 73%) | $1.58 \times 10^2$ (20/30; 66%) | $1.60 \times 10^2$ (20/30; 67%) | $1.33 \times 10^2$ (20/30; 67%) |
| | TADV-A$_{Hm}$ | $0.00 \times 10^1$ (0/30; 0%) | $0.00 \times 10^1$ (0/30; 0%) | $0.00 \times 10^1$ (0/30; 0%) | $0.00 \times 10^1$ (0/30; 0%) |
| DENV-4 | WT | $3.25 \times 10^3$ (25/30; 83%) | $3.80 \times 10^4$ (25/32; 78%) | $1.50 \times 10^5$ (25/32; 78%) | $1.60 \times 10^5$ (25/32; 78%) |
| | TADV-A$_{Ht}$ | $3.40 \times 10^1$ (22/30; 73%) | $2.38 \times 10^2$ (19/28; 68%) | $2.95 \times 10^2$ (19/28; 68%) | $1.08 \times 10^2$ (19/28; 68%) |
| | TADV-A$_{Hm}$ | $0.00 \times 10^1$ (0/27; 0%) | $0.00 \times 10^1$ (0/28; 0%) | $0.00 \times 10^1$ (0/28; 0%) | $0.00 \times 10^1$ (0/28; 0%) |

homozygous and heterozygous mosquitoes at 14 dpi. The results from three biological replicates indicated that none of the homozygous TADV-A mosquitoes (n = 30) were positive for viral replication (dissemination) in either the midgut or the midgut-free carcass (Fig 1B, Table 1). DENV-2 prevalence was detected in 86.6% (26/30) of heterozygous TADV-A mosquitoes in both the carcass and midgut; however, they also had significantly ($p < 0.001$) lower levels of viral RNA (~3 log$_{10}$) compared to the WT (Fig 1B, Table 1). Finally, as transmission occurs through the saliva, viral transmission rates were determined by collecting the saliva from individual mosquitoes at 14 dpi and measuring the DENV-2 titers using an assay for the median tissue culture infective dose (TCID$_{50}$). No DENV-2 was detected in the saliva of homozygous TADV-A mosquitoes (n = 30) (Fig 1B), though it was detected in 83.3% (25/30) of heterozygous TADV-A mosquitoes; however, here again the DENV-2 titers were significantly ($p < 0.001$) lower (3.56 x 10$^2$ TCID$_{50}$/ml/mosquito) than the WT mosquitoes (2.70 x 10$^5$ TCID$_{50}$/ml/mosquito) (Fig 1B, Table 1).

To determine whether the anti-DENV 1C19 scFv is broadly inhibitory for other DENV serotypes, the vector competence of TADV-A mosquitoes was assessed using DENV-1 (isolate ET243, GenBank EF440432), DENV-3 (isolate ET209, Genbank EF440434), and DENV-4 (isolate ET288, Genbank EF440435). Tests for infection, dissemination, and transmission were

carried out as above, and the results, presented together in Fig 1B and Table 1, were comparable to those obtained with the DENV-2 serotype. In short, the TADV-A mosquitoes homozygous for the transgene proved to be refractory to infection with all three additional serotypes also showing no infection in their midguts at 4 dpi (DENV-1 n = 28; DENV-3 n = 30; DENV-4 n = 27). Even at 14 dpi, there was no sign of viral replication in the midgut or carcass for all tested specimens, and none of the saliva samples (DENV-1 n = 28; DENV-3 n = 30; DENV-4 n = 28) were positive for the virus. As with DENV-2, the mosquitoes heterozygous for the transgene still tested positive for the virus in most specimens, though the overall DENV titers were significantly lower than compared to the WT in all cases (Fig 1B; Table 1).

## Engineered anti-DENV mosquitoes outperform *Wolbachia*

To compare the inhibitory effect of the anti-DENV 1C19 scFv to DENV inhibition through *Wolbachia*[65–67] infection, we challenged WT mosquitoes infected with *Wolbachia* (*w*Mel) with DENV-2. Vector competence results revealed that midguts from mosquitoes infected with *Wolbachia* had significantly ($p < 0.001$) reduced DENV-2 RNA levels ($4.75 \times 10^1$ GE) at 4 dpi compared with the WT ($4.25 \times 10^4$ GE) (Fig 1B, Table 1). Similarly, viral dissemination at 14 dpi was also reduced ($p < 0.001$) in *w*Mel mosquitoes ($\sim 3 \log_{10}$), and DENV titers in mosquito saliva at 14 dpi were significantly ($p < 0.01$) lower ($\sim 3 \log_{10}$) in *w*Mel mosquitoes ($4.90 \times 10^1$ TCID$_{50}$/ml/mosquito) than in the WT ($2.70 \times 10^5$ TCID$_{50}$/ml/mosquito) (Fig 1B, Table 1). Importantly, a direct comparison revealed that the TADV-A mosquitoes are significantly ($p < 0.001$) more effective as homozygotes, and similarly effective as heterozygotes, at blocking DENV infection as *Wolbachia*-infected mosquitoes.

## Transgene impact on fitness

To determine whether the anti-DENV 1C19 scFv had any significant fitness effects on transgenic mosquitoes, we assessed several fitness parameters including larval to pupal development time, male and female fecundity and fertility, male mating success, and longevity (Table 2). No significant differences were observed between WT and TADV-A mosquitoes when examining male mating success and fecundity and fertility in both males and females ($p > 0.05$). However, we noticed a significant difference in larval to pupal development times ($p < 0.0001$), with WT mosquitoes developing, on average, 0.8 days faster than TADV-A mosquitoes. When assessing mosquito survivorship, there was no significant difference between WT and TADV-A males ($p > 0.05$; S4 Fig), though WT female mosquitoes lived, on average, 4.5 days longer than TADV-A females ($p < 0.05$; S4 Fig). The longevity of infected mosquitoes was also assessed. Transgenic, WT, or *w*Mel mosquitoes were infected with four DENV serotypes and their survivorship was assessed 14 dpi (Table 2). No significant ($p > 0.01$) differences between WT and TADV-A longevity upon infection with serotypes DENV-2, -3, and -4 were observed. However, there was a significant difference in survival upon infection with serotype DENV-1, with a higher proportion of WT mosquitoes surviving at 14 dpi ($p < 0.01$; Table 2, S4 Fig). In addition, a significant difference in survival between *w*Mel mosquitoes and WT and TADV-A mosquitoes infected with serotype DENV-2 was observed ($p < 0.0001$; S4 Fig).

## Discussion

Our results demonstrate that conditional expression of the anti-DENV 1C19 scFv renders mosquitoes refractory to all four major DENV serotypes and therefore appears to be a potent viral inhibition strategy. While mosquitoes homozygous for the anti-DENV 1C19 scFv showed complete refractoriness to DENV infection, heterozygous mosquitoes were still partially refractory to DENV infection, dissemination, and transmission, with significant, several

**Table 2. Effect of anti-DENV scFv on fitness.** Comparisons of several fitness parameters (left-most column) between WT (second column from left) and TADV-A mosquitoes (third column from left) suggest that there are few significant differences (right-most column) between the two groups, indicating that the anti-DENV scFv does not have a major impact on mosquito fitness. The survivorship of infected and non-infected mosquitoes is also shown. The median survival in days was determined for non-infected mosquitoes, and the percent of surviving mosquitoes separately infected by four DENV serotypes was assessed at 14 dpi.

| | Strain | | |
| --- | --- | --- | --- |
| Fitness Parameter | WT (N) | TADV-A (N) | *P*-value |
| Female fecundity$\theta^{\dagger\S}$ | 103.6 ± 3.8 (60; 6,213) | 110.2 ± 4.4 (57; 5,756) | 0.2578 |
| Egg hatchability$\theta^{\|\S}$ | 67.5 ± 3.2 (60; 4,208) | 61.0 ± 4.1 (57; 4,046) | 0.2149 |
| Male mating success$\theta^{o\P}$ | 93.00 ± 0.04 (43) | 95.00 ± 0.04 (37) | 0.7756 |
| Male fecundity$\theta^{lc}$ | 226.3 ± 15.7 (43; 9,730) | 202.7 ± 17.2 (37; 7,318) | 0.3141 |
| Egg hatchability$\theta^{III\S}$ | 75.9 ± 4.5 (43; 7,558) | 73.1 ± 3.9 (37; 5,624) | 0.6260 |
| Larval to pupal development in days$\theta^{\S}$ | 6.70 ± 0.77 (1,322) | 7.50 ± 0.09 (774) | <0.0001 |
| Female median survival in days[††] | 53 (122) | 48.5 (128) | 0.0129 |
| Male median survival in days[††] | 14 (175) | 14 (184) | 0.1781 |
| % Survival at 14 dpi with DENV-1[‡††] | 80.8 (26) | 43.5 (23) | 0.0086 |
| % Survival at 14 dpi with DENV-2[‡††] | 72.7 (33) | 69.2 (34) | 0.6891 |
| % Survival at 14 dpi with DENV-3[‡††] | 64.9 (37) | 52.2 (46) | 0.2679 |
| % Survival at 14 dpi with DENV-4[‡††] | 41.3 (138) | 48.8 (41) | 0.7256 |

$\theta$Mean ± SEM reported.

[†]Average number of eggs laid per female (Number of females scored; total number of eggs counted).

[‖]Percentage of laid eggs that produced larvae (Number of females scored; total number of larvae counted).

[o]Percentage of single male outcrosses that gave rise to viable progeny.

[l]Average number of eggs laid per single male outcross (Number of male outcrosses scored; total number of eggs counted).

[III]Percentage of laid eggs that produced larvae per single male outcross (Number of male outcrosses scored; total number of larvae counted).

[§]Unpaired t test with Welch's correction was used.

[¶]Unpaired t test was used to evaluate the statistical significance between the proportions of fertile males.

[††]Mantel-Cox test was used.

[‡]Percentage of infected mosquitoes surviving at 14 dpi.

orders-of-magnitude reductions in viral titers in the saliva. Given previous characterizations of the 1C19 scFv antibody, we presume that it achieves this refractoriness because, when it is secreted into the epithelium of the posterior midgut in mosquitoes [63], it binds to the exposed fusion loop of DENV and inhibits the virus particle from releasing its genome into the cytoplasm for replication. Based on previous findings, it is likely that this significant reduction in viral titers would be sufficient to render heterozygous mosquitoes unable to transmit DENV to a susceptible host[68]. Though this remains to be demonstrated, our results show that heterozygous 1C19 scFv antibody-expressing transgenic mosquitoes are just as efficient at viral suppression as—and homozygous mosquitoes are significantly more efficient than—*Wolbachia*-infected mosquitoes, which are currently being released for DENV control because they are known to be refractory to DENV[65].

The difference in refractory levels in the homozygous versus heterozygous mosquitoes also suggests that the refractory phenotype is particularly sensitive to scFv antibody expression levels, a phenomenon previously observed with anti-malarial scFv transgenes [41] and anti-ZIKV synthetic small RNA transgenes [69]. If this means that complete refractoriness is susceptible to positional effects, e.g., not refractory when the scFv antibody transgene is expressed from a different, possibly more weakly expressing genomic insertion position, the identification of more robust midgut-specific promoters may help to ensure sufficiently high expression levels from a single copy of the transgene regardless of insertion site, as can the use of multiple anti-

DENV scFv antibodies in a single transgene [41,43]. Additionally, while we observed no significant reduction in multiple fitness parameters in transgenic mosquitoes when compared to WT, some differences in fitness were observed, and more extensive analyses on fitness of both infected and uninfected transgenic heterozygotes and homozygotes (possibly after introgression with a field-collected mosquito strain) would have to be performed before use of such mosquitoes in the field.

The strategy we describe here provides an efficient "cargo" gene that can be coupled with a gene-drive system to reduce or eliminate the risk of DENV transmission by mosquitoes. In fact, previous efforts have demonstrated effective Cas9-mediated homing-based gene drives in malaria vectors [70–72], and even *Ae. aegypti* [34]. Additionally, since homing-based drive systems quickly convert heterozygotes to homozygotes [25], linking the anti-DENV 1C19 scFv antibody described here to such a drive system could, in theory, rapidly convert wild mosquito populations into transgenic homozygotes that would be completely resistant to DENV transmission. Of paramount importance to the viability of such an approach is the evolutionary stability of the refractory transgene, specifically in terms of the likelihood of viral-resistant evolution. Indeed, several studies have shown that, in some contexts, DENV can rapidly evolve resistance in response to neutralizing antibodies [73,74], and this may be especially likely in the TADV-A heterozygotes described in this study. However, this potentially can be managed through the selection of antibodies with mechanisms/epitopes that minimize the chance of evolved resistance and the use of a combination of distinct anti-DENV antibodies (many of which have been described, e.g.,[44,73–76]), as well as a combination of antibody and non-antibody based DENV refractoriness transgenes (e.g., [36,37]; [38]; [39])). The deployment of such a pan-serotype-targeting strategy could serve as an effective component of a comprehensive program to reduce the incidence and impact of DENV.

Due to similarities within viral families, this research could have far-reaching consequences for rendering mosquitoes resistant to other arboviruses like ZIKV and CHIKV by using similar genetic engineering strategies to develop scFv-based transgenes. Multiple potent antibodies that effectively neutralize these various mosquito-borne viruses have also been identified in the last decade[77–81]. Although not all of these will confer robust viral resistance when expressed *in vivo* in mosquitoes, the availability of diverse, well-characterized antibodies of this sort, largely as a result of antibody therapeutic development efforts[78], should allow for the identification of those that function within the desired context. Given the increasing incidence of disease caused by these viruses and the resulting global health implications, such scFv-based transgenes coupled with gene-drive systems[34] can provide an effective, sustainable, and comprehensive strategy for reducing the impact of arboviral mosquito-borne diseases.

## Supporting information

**S1 Fig. Expression correlation analyses of gene expression levels (indicated by TPM [transcripts per million] values) in dissected midgut tissues from WT or transgenic (TADV-A, -B, or -C) mosquitoes without a blood meal (A) and 24 hours after a blood meal (B).** The y-axis corresponds to TPM values in WT samples, and the x-axis corresponds to TPM values in respective transgenic samples. Blue-colored points represent endogenous genes, and red-colored points represent the 1C19 scFv. Comparisons between WT and TADV samples suggest that 1C19 scFv expression is transgene-dependent and does not appear to significantly affect global expression levels of endogenous RNAs. Pearson correlation coefficients ($r$) between gene expression levels in WT versus transgenic samples are reported in bold in their respective graphs.
(TIF)

**S2 Fig. Western blot analyses to probe for the presence of the 1C19 scFv antibody protein in WT and transgenic midgut samples.** Western blots were carried out utilizing a custom antibody specific for the 1C19 scFv protein, as well as an antibody to recognize the 3xHA tag, on dissected midgut tissues from non-blood-fed or 16-hour post-blood-meal WT or TADV-A, TADV-B, or TADV-C mosquitoes. An anti-GAPDH antibody was used as a control. The presence of a 26.3-kDa band confirms the expression of the 1C19 scFv protein in transgenic, but not in WT, mosquito midgut samples. The presence of a 30 kDa band indicates the presence of the 3xHA tag in TADV-B and TADV-C but not in WT or TADV-A mosquitoes.
(TIF)

**S3 Fig. Effect of the anti-DENV scFv on DENV GE in three independent transgenic mosquito lines.** DENV GE in WT and transgenic mosquito lines (TADV-A, TADV-B, or TADV-C) following a blood meal infected with DENV-2 (ET300 strain) are shown. DENV GE from mosquito midguts (at 4 dpi) of WT or transgenic mosquitoes were determined using real-time qPCR and calculated using previously published methods. Circles represent WT mosquitoes; black diamonds represent anti-DENV homozygous transgenic mosquitoes; red colored diamonds represent anti-DENV heterozygous transgenic mosquitoes. Horizontal bars represent the mean viral titer. The Mantel-Cox test was used for statistical analysis. $^{**}p < 0.001$.
(TIF)

**S4 Fig. Survivorship curves for uninfected WT and TADV-A mosquitoes and for DENV-infected WT, _w_Mel, and TADV-A mosquitoes.** The x-axis indicates the number of days elapsed after the start of the experiment, and the y-axis indicates the percent of mosquitoes surviving on each elapsed day. (**A**) For the uninfected panel, where the survivorship curves for WT and TADV-A male and female mosquitoes are shown separately, each line represents the accumulated results of 120–180 adult mosquitoes combined from 3 biological replicates. Significant differences in survivorship were observed between WT and TADV-A females, with WT females surviving, on average, 4.5 days longer ($p \leq 0.01$). (**B**) For the infected panels, WT, _w_Mel, and TADV-A females were given a blood meal infected with DENV-1, DENV-2, DENV-3, or DENV-4 (as indicated on respective plot titles). The survivorship of infected mosquitoes was determined over the course of 14 days (the time it takes for the virus to disseminate past the midgut and eventually become transmissible). No significant differences in survivorship were found between WT and TADV-A mosquitoes when infected with DENV-3 and DENV-4, but significant differences were observed upon infection with DENV-1, with more WT mosquitoes (80%) surviving at 14 dpi than TADV-A mosquitoes (~40%; $p \leq 0.01$). When infected with DENV-2, more _w_Mel mosquitoes (>90%) survived at 14 dpi compared to WT and TADV-A mosquitoes (both ~70%; $p < 0.0001$). The Mantel-Cox test was used to determine statistical significance. $^{*}p \leq 0.01$, $^{***}p < 0.0001$.
(TIF)

**S1 Table. Effect of the anti-DENV scFv on DENV-2 GE in three independent mosquito lines.** DENV-2 GE are shown below for WT, TADV-A, TADV-B, and TADV-C mosquito lines following a blood meal infected with the DENV-2 ET300 strain. Midgut samples were collected 4 dpi, and GE were determined using real-time RT-qPCR and calculated using previously published methods.
(DOCX)

**S2 Table. Primers utilized to generate anti-DENV scFv used in this study.**
(DOCX)

**S3 Table. Diagnostic primers used for inverse PCR (iPCR) assays, zygosity confirmation, pan-DENV serotype detection, and *w*Mel infection confirmation.**
(DOCX)

**S4 Table. Quantification of total RNA expression in WT, TADV-A, TADV-B, and TADV-C mosquito midguts prior to blood meal (NBF) and 24-hours post-blood feeding (PBM).** Both raw read counts and normalized TPM values are included.
(XLSX)

**S5 Table. Differential RNA expression by a two-factor DESeq analysis in WT, TADV-A, TADV-B, TADV-C mosquito midguts prior to blood meal (NBF) and 24-hours post-blood feeding (PBM).** Statistics such as p-value and p-adj are included. Gene expression changes (in terms of log2FoldChange) describe the genes that are upregulated in either the WT (negative values) or transgenic mosquitoes (positive values).
(XLSX)

**S6 Table. List of genes showing consistently altered expression in transgenic (TADV-A, TADV-B, and TADV-C) versus WT mosquito midguts.** Only 9 genes were found to be consistently upregulated in all three transgenic lines when compared to WT.
(XLSX)

## Author Contributions

**Conceptualization:** Prasad N. Paradkar, Omar S. Akbari.

**Data curation:** Anna Buchman, Stephanie Gamez, Ming Li, Igor Antoshechkin, Hsing-Han Li, Hsin-Wei Wang, Melissa J. Klein, Jean-Bernard Duchemin, Prasad N. Paradkar.

**Formal analysis:** Anna Buchman, Igor Antoshechkin, Hsin-Wei Wang, Chun-Hong Chen, Melissa J. Klein.

**Funding acquisition:** Omar S. Akbari.

**Investigation:** Stephanie Gamez, Ming Li, Hsing-Han Li, Hsin-Wei Wang, Chun-Hong Chen, Melissa J. Klein, James E. Crowe, Jr., Prasad N. Paradkar.

**Methodology:** James E. Crowe, Jr., Prasad N. Paradkar.

**Resources:** James E. Crowe, Jr., Prasad N. Paradkar.

**Writing – original draft:** Anna Buchman, Igor Antoshechkin, Prasad N. Paradkar, Omar S. Akbari.

**Writing – review & editing:** Anna Buchman, Stephanie Gamez, Igor Antoshechkin, Chun-Hong Chen, Prasad N. Paradkar, Omar S. Akbari.

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
