## [Decision Letter · Decision Letter 0]

28 Aug 2019

Dear Dr. Akbari:

Thank you very much for submitting your manuscript "Broad Dengue Neutralization in Mosquitoes Expressing an Engineered Antibody" (PPATHOGENS-D-19-01283) for review by PLOS Pathogens. Your manuscript was fully evaluated at the editorial level and by independent peer reviewers. The reviewers appreciated the attention to an important topic but identified some aspects of the manuscript that should be improved.

We therefore ask you to modify the manuscript according to the review recommendations before we can consider your manuscript for acceptance. Your revisions should address the specific points made by each reviewer.

(1) A letter containing a detailed list of your responses to the review comments and a description of the changes you have made in the manuscript. Please note while forming your response, if your article is accepted, you may have the opportunity to make the peer review history publicly available. The record will include editor decision letters (with reviews) and your responses to reviewer comments. If eligible, we will contact you to opt in or out.

(2) Two versions of the manuscript: one with either highlights or tracked changes denoting where the text has been changed; the other a clean version (uploaded as the manuscript file).

We hope to receive your revised manuscript within 60 days or less. If you anticipate any delay in its return, we ask that you let us know the expected resubmission date by replying to this email.

[LINK]

Sincerely,

Mehul Suthar

Associate Editor

PLOS Pathogens

Ana Fernandez-Sesma

Section Editor

PLOS Pathogens

Kasturi Haldar

Editor-in-Chief

PLOS Pathogens

orcid.org/0000-0001-5065-158X

Grant McFadden

Editor-in-Chief

PLOS Pathogens

orcid.org/0000-0002-2556-3526

Reviewer's Responses to Questions

**Part I - Summary**

Reviewer #1: The authors present new genetically engineered Ae. aegypti strain expressing a highly and broadly cross-neutralizing single chain antibody against DENV. They demonstrate through a series of experiments that these engineered mosquitoes can effectively (100%) inhibit the infection, dissemination and transmission of all four serotypes of DENV. Further, they demonstrate that their strain is even more effective than Wolbachia, which is currently the gold standard for DENV inhibition. Overall this is a very exciting and impactful manuscript. Below are my critiques.

Reviewer #2: The study by Buchman et al generated Aedes aegypti mosquitoes that produce a single chain antibody that is capable of neutralizing all four strains of Dengue virus. The paper is well written and contains necessary controls and experiments for the conclusions they are drawing. They engineered the antibody expression to coincide with blood meal digestion to limit expression and find it had little impact on mosquito fitness. Mosquitoes homozygous for the antibody appear completely refractory for virus whereas heterozygotes tested positive for virus, but contained significantly lower quantities of virus. They include Wolbachia infected mosquitoes to compare their results with other well-established strategies to decrease mosquito competence for Dengue transmission. They find that the heterozygotes carrying the antibody decrease viral quantities similar to Wolbachia infected mosquitoes and homozygotes are more effective. While outside of this particular study, I cannot help but wonder how quickly Dengue will be able to escape neutralization from a single antibody. This is a very provocative study demonstrating a new method to decrease dengue spread. Future work would require incorporating antibody expression with a potent gene drive.

Reviewer #3: This is a very elegant study where transgenic Aedes aegypti expressing an engineered single-chain variable fragment derived from a broadly neutralizing dengue virus human monoclonal antibody are refractory to all four dengue virus serotypes. As such the study can be considered a significant advance that paves the way to develop transgenic mosquito-based dengue control. One weakness is the lack of basic biological insight.

Reviewer #4: Strengths: This paper by Buchman et al builds on previous work by the group where they engineered miRNAs in Aedes aegypti for resistance to DENV1-4. In the paper under review, they have engineered Mab 1C19 as a single-chain variable fragment (scFv) to build an effector transgene for resistance to multiple DENV serotypes. The key innovative aspect of this work is the development of the scFv from IC19 since the monoclone has broad neutralzing activity against the four serotypes. This type of approach (targeting all four serotypes) has been a 'holy grail' of geneticists/arbovirologists interested in designing Aedes aegypti for resistance to DENVs and furthers population replacement strategies/approaches. This research group also has a continued interest in developing antiviral transgenes in the context of gene drive so this might all come together. An important feature of this work is the IC19 scFv expression levels are increased with homozygosity generating apparent complete resistance to DENV1-4. It was a nice touch to compare DENV resistance of TADV-A vs wMel Wolbachia infected mosquitoes. The paper is well written and follows a logical experimental approach for developing transgenesis and characterizing transgene efficiency and a DENV 1-4 resistance phenotype.

Weaknesses: The weaknesses are minor. However, the approach brings up two points. First, in theory minor changes in the viral genome at the target site could reduce or ablate resistance by altering the ability of the scFv to neutralize. The authors discussed this in their paper and thought they could manage this outcome through selection of other scFVs transgenes, but these may run into the same problem. Nevertheless this approach may drive evolution/selection of viruses that fail to interact with these transgene effectors. Perhaps building transgenics with multiple transgenes may mitigate the ability of DENVs to adapt to any one transgene and designing a mosquito with a combination of effector gene approaches may be a future goal. Second, the fitness of the TADV-A may be further improved by introgressing the scFv transgene into a field-collected (true wild-type) strain of Aedes aegypti since the progenitor HWE strain contains recessive lethals that compromise fitness. However, I view these as minor issues for this paper. Please address.

**Part II – Major Issues: Key Experiments Required for Acceptance**

Reviewer #3: The authors could have elaborated more on biological aspects, such as the RNAseq data (unlikely that no endogenous mosquito gene were not regulated). The authors should also elaborate on the fitness data that did show some interesting patterns especially for dengue virus 1.The comparison between the transgenic mosquitoes and Wolbachia only involved one virus strain (a lab strain) so one should be cautious with the interpretation of that data.

**Part III – Minor Issues: Editorial and Data Presentation Modifications**

Reviewer #1: 1. The following excerpt is a run-on sentence and should be changed. Line numbering would have been helpful.

“Moreover, releases of mosquitoes artificially infected with the intracellular endosymbiont Wolbachia, which can make infected males incapable of successfully mating with uninfected females in an SIT-like manner and can inhibit mosquito infection with pathogens such as DENV and ZIKV(Aliota et al., 2016;Walker et al., 2011), have also been carried out to either suppress mosquito populations or make them less likely to transmit pathogens, and may hold promise for reducing incidence of disease(Moreira et al., 2009; Schmidt et al., 2017).”

2. A description of the carboxypeptidase promoter should be presented. Presumably, it is constitutively expressed and only translated following a bloodmeal? This should be included somewhere in the manuscript.

3. It is unclear where this scFv is expressed. Is it secreted, membrane bound or cytosolic? There was no mention of its localization. If it is secreted how does the antibody bind the virus prior to infection? Presumably, transcripts will be translated upon bloodfeeding (see above comment), but temporally how does this align with the rate of midgut infection? Can the antibodies still block infection after internalization into early endosomes? I am not suggesting this paper needs to be converted into a mechanistic paper, but it should discuss these questions.

4. The first paragraph of the discussion discusses how heterozygous mosquitoes could be sufficient to halt transmission due to their reduction in transmission. This may or may not be true which is acknowledged; however, if this argument that heterozygous mosquitoes could be effectively used in nature if needed than it would have been nice to see the fitness studies expanded to include the TADV-A heterozygotes.

5. The authors discuss the possibility of escape mutants in the discussion. I don’t think this is a major concern in the homozygotes because they demonstrated sterilizing immunity; however, this could be important in heterozygotes.

Reviewer #2: Minor:

What was the concentration of virus in the blood-meal? I was unable to find this information in the manuscript.

Sup Fig 1: Western blots for the Non blood-fed midguts should be run on the same gel as the positives to demonstrate relative expression levels. Running the samples on multiple gels makes things harder to evaluate and potentially makes the background bands more apparent.

PLOS authors have the option to publish the peer review history of their article (what does this mean?). If published, this will include your full peer review and any attached files.

Reviewer #1: No

Reviewer #2: No

Reviewer #3: No

Reviewer #4: No

---

## [Editor Report · Decision Letter 1]

23 Sep 2019

Dear Dr. Akbari,

We are pleased to inform that your manuscript, "Broad Dengue Neutralization in Mosquitoes Expressing an Engineered Antibody", has been editorially accepted for publication at PLOS Pathogens. 

Before your manuscript can be formally accepted and sent to production, you will need to complete our formatting changes, which you will receive by email within a week. Please note that your manuscript will not be scheduled for publication until you have made the required changes.

IMPORTANT NOTES

(1) Please note, once your paper is accepted, an uncorrected proof of your manuscript will be published online ahead of the final version, unless you’ve already opted out via the online submission form. If, for any reason, you do not want an earlier version of your manuscript published online or are unsure if you have already indicated as such, please let the journal staff know immediately at plospathogens@plos.org.

(2) Copyediting and Proofreading: The corresponding author will receive a typeset proof for review, to ensure errors have not been introduced during production. Please review the PDF proof of your manuscript carefully, as this is the last chance to correct any errors. Please note that major changes, or those which affect the scientific understanding of the work, will likely cause delays to the publication date of your manuscript. 

(3) Appropriate Figure Files: Please remove all name and figure # text from your figure files. Please also take this time to check that your figures are of high resolution, which will improve the readbility of your figures and help expedite your manuscript's publication. Please note that figures must have been originally created at 300dpi or higher. Do not manually increase the resolution of your files. For instructions on how to properly obtain high quality images, please review our Figure Guidelines, with examples at: http://journals.plos.org/plospathogens/s/figures.

(4) Striking Image: Please upload a striking still image to accompany your article if one is available (you can include a new image or an existing one from within your manuscript). Should your paper be accepted, this image will be considered for our monthly issue image and may also appear on our website to feature your article. Please upload this as a separate file, selecting "striking image" as the file type upon upload. Please also include a separate "Other" file with a caption, including credits and any potential copyright information. Please do not include the caption in the main article file. If your image is from someone other than yourself, please ensure that the artist has read and agreed to the terms and conditions of the Creative Commons Attribution License at http://journals.plos.org/plospathogens/s/content-license. Please note that PLOS cannot publish copyrighted images.

(5) Press Release or Related Media: If your institution or institutions have a press office, please notify them about your upcoming paper at this point, to enable them to help maximize its impact. If they will be preparing press materials for this manuscript, please inform our press team in advance at plospathogens@plos.org as soon as possible. We ask that you contact us within one week to plan ahead of our fast Production schedule. If you need to know your paper's publication date for related media purposes, you must coordinate with our press team, and your manuscript will remain under a strict press embargo until the publication date and time. This means an early version of your manuscript will not be published ahead of your final version. 

(6)  PLOS requires an ORCID iD for all corresponding authors on papers submitted after December 6th, 2016. Please ensure that you have an ORCID iD and that it is validated in Editorial Manager.  To do this, go to ‘Update my Information’ (in the upper left-hand corner of the main menu), and click on the Fetch/Validate link next to the ORCID field.  This will take you to the ORCID site and allow you to create a new iD or authenticate a pre-existing iD in Editorial Manager

(7) Update your Profile Information: Now that your manuscript has been provisionally accepted, please log into Editorial Manager and update your profile, if needed. Go to https://www.editorialmanager.com/ppathogens, log in, and click on the "Update My Information" link at the top of the page. Please update your user information to ensure an efficient production and billing process. 

(8) LaTeX users only: Our staff will ask you to upload a TEX file in addition to the PDF before the paper can be sent to typesetting, so please carefully review our Latex Guidelines http://journals.plos.org/plospathogens/s/latex in the meantime.

(9) If you have associated protocols in protocols.io, please ensure that you make them public before publication to guarantee immediate access to the methodological details.

Best regards,

Mehul Suthar

Associate Editor

PLOS Pathogens

Ana Fernandez-Sesma

Section Editor

PLOS Pathogens

Kasturi Haldar

Editor-in-Chief

PLOS Pathogens

orcid.org/0000-0001-5065-158X

Grant McFadden

Editor-in-Chief

PLOS Pathogens

orcid.org/0000-0002-2556-3526
---

## [Editor Report · Acceptance letter]

6 Dec 2019

Dear Dr. Akbari,

We are delighted to inform you that your manuscript, "Broad Dengue Neutralization in Mosquitoes Expressing an Engineered Antibody," has been formally accepted for publication in PLOS Pathogens.

Best regards,

Kasturi Haldar

Editor-in-Chief

PLOS Pathogens

orcid.org/0000-0001-5065-158X

Grant McFadden

Editor-in-Chief

PLOS Pathogens

orcid.org/0000-0002-2556-3526